# The Current Genomic and Molecular Landscape of Philadelphia-like Acute Lymphoblastic Leukemia

**DOI:** 10.3390/ijms21062193

**Published:** 2020-03-22

**Authors:** Parveen Shiraz, Kimberly J. Payne, Lori Muffly

**Affiliations:** 1Division of Blood and Marrow Transplantation, Stanford University, Stanford, CA 94305, USA; lmuffly@stanford.edu; 2Department of Pathology and Human Anatomy, Loma Linda University, Loma Linda, CA 92350, USA; kpayne@llu.edu

**Keywords:** acute lymphoblastic leukemia, CRLF2, PI3K/AKT/mTOR, targeted therapy

## Abstract

Philadelphia (Ph)-like acute lymphoblastic leukemia (ALL) is a high-risk B-cell Acute Lymphoblastic Leukemia (B-ALL) characterized by a gene expression profile similar to Ph-positive B-ALL but lacking the *BCR-ABL1* translocation. The molecular pathogenesis of Ph-like B-ALL is heterogenous and involves aberrant genomics, receptor overexpression, kinase fusions, and mutations leading to kinase signaling activation, leukemogenic cellular proliferation, and differentiation blockade. Testing for the Ph-like signature, once only a research technique, is now available to the clinical oncologist. The plethora of data pointing to poor outcomes for this ALL subset has triggered investigations into the role of targeted therapies, predominantly involving tyrosine kinase inhibitors that are showing promising results.

## 1. Introduction

Tremendous progress has been achieved in childhood acute lymphoblastic leukemia (ALL) with long-term survival currently approaching 90% [1]. Outcomes in adult ALL have also improved, albeit, not to the same extent, with long-term survival ranging from 40–60% depending on the age group, ALL subtype, and the regimen used [2,3]. Retrospective and prospective data have demonstrated improved outcomes with pediatric inspired regimens in the adolescent and young adult (AYA) ALL population [2,4,5,6,7]. However, B-ALL with the Philadelphia-like signature (Ph-like ALL) stands out as a distinct high-risk entity despite the use of pediatric inspired regimens [2]. Ph-like ALL, also termed *BCR-ABL1*-like ALL, is a subset of B-ALL defined by the presence of a gene expression profile similar to Ph-positive ALL but without the t(9;22) or *BCR-ABL1* translocation. It is characterized by a heterogenous genetic and molecular landscape that includes chromosomal translocations, loss of tumor suppressor genes, kinase gene fusions and mutations, kinase pathway activation, and dysregulated lymphoid transcription factors. Ph-like ALL is recognized as a provisional entity in the 2016 revision to the World Health Organization (WHO) classification of myeloid neoplasms and acute leukemia [8]. Ph-like ALL comprises approximately 15% of B-ALL in children [9,10,11,12,13], increases to 20–27% B-ALL in the AYA population [14], and 20% of B-ALL in older adults [15,16], with the highest incidence being in the AYA population. This entity almost always occurs in the subgroup of ALL without *BCR-ABL1*, *ETV6-RUNX1*, *MLL/KMT2A*, and *TCF3* rearrangements [12]. In this review, we will describe the genomic landscape, molecular aberrations, and potential therapeutic avenues for treatment of Ph-like ALL.

## 2. Initial Description and Discovery of Ph-like B-ALL

In the early 2000s, it was recognized that chromosomal abnormalities alone were insufficient to initiate leukemogenesis in ALL and did not fully account for the poor treatment outcomes in certain ALL subsets. This led to a search for other genetic alterations in childhood and adult ALL. The late 1990s and early 2000s also saw rapid progress and automation of DNA microarray technology. This enabled many investigator groups to study the gene expression profiles of different ALL subsets [17,18,19,20]. However, it was not until 2009 that the molecular and clinical description of Ph-like B-ALL was reported simultaneously by two independent groups of investigators: St. Jude’s/Children’s Oncology Group (COG) in the United States (US) and the Dutch Childhood Oncology Group (DCOG). These two studies reported a similar gene expression profile as Ph-positive ALL but without the hallmark *BCR-ABL1* translocation [12,21]. Although their methodologies differed (Table 1), both groups clearly demonstrated the biological complexity and poor clinical outcomes associated with Ph-like ALL in children.

The COG investigators led by Mullighan et al. [21] identified a group of high-risk B-ALL from the COG P9906 study and used the Affymetrix single-nucleotide-polymorphism (SNP) array analysis with 257 probe sets to perform gene-expression profiling. Deletions in *CDKN2A/B*, *PAX5*, *IKZF1*, *ETV6*, *RB1*, and *BTG1* were commonly detected. Supervised principal-components analysis was used to associate gene copy number and treatment outcome. Among the genetic aberrations detected, only *IKZF1* alteration was associated with poor outcomes. The 10-year incidence of relapse was ~50% in the *IKZF1* altered cases compared to ~20% in the *IKZF1* intact cases. 

The Dutch Group led by Den Boer et al. studied all risk groups of newly diagnosed childhood ALL from the German Cooperative ALL (COALL) 92/97 and DCOG-ALL-8/9 studies [12]. They used the Affymetrix U133 GeneChips for RNA hybridization. Hierarchical clustering with 110 gene-probe sets revealed that 15% of B-ALL cases co-clustered with *BCR-ABL1-*positive cases but lacked the *BCR-ABL1* translocation, yet had similar poor outcomes with relapse rates of approximately 50%. In vitro drug toxicity assays revealed the *BCR-ABL*-like leukemic cells were 73 times more resistant to l-asparaginase and 1.6 times more resistant to daunorubicin when compared to other precursor B-ALL cases. No differences were seen in the sensitivity to prednisolone and vincristine. Differences between the two cohorts, methodologies, and outcomes are summarized in Table 1. Since the initial description, other groups have identified similar genomic profiles in additional cohorts of Ph-like ALL in both children and adults [10,22,23]. 

## 3. Clinical Features and Outcomes

Multiple studies have identified poor outcomes for patients with Ph-like B-ALL across all age groups. Ph-like B-ALL is associated with a higher risk of induction failure, measurable residual disease (MRD) positivity [24], higher relapse rates (as high as 70% at 3 years), and lower overall survival [4,9,10,12,13,14,16,22,25,26,27,28,29,30,31]. In the initial description of Ph-like B-ALL in pediatric patients reported by investigators from St. Jude’s Children’s Research Hospital, deletion or mutation of *IKZF1* was associated with poor outcomes. The 10-year incidence of relapse was 48% and 25% in the *IKZF1*-altered and unaltered groups, respectively (*p* = 0.004). High level MRD at day 29 was significantly more frequent in the *IKZF1*-altered group at 23% compared to 6% in the *IKZF1*-intact group (*p* = 0.001%) [21]. In a retrospective analysis of pediatric patients enrolled in the COG AALL0331 between 2006 and 2008, the Ph-like gene signature was identified in 13% of National Cancer Institute (NCI) standard risk B-ALL, and these patients had inferior 7-year event-free survival (EFS) compared to those without the Ph-like gene signature (82% versus 90%, *p* = 0.0022). However, there was no difference in overall survival (OS, 93% versus 95%, *p* = 0.14) [9].

Data compiled from multiple cooperative group studies of children, adolescents, and young adults with ALL reveal that for Ph-like ALL subgroups, children have better outcomes compared to adolescents who in turn fare better than young adults. The median 5-year EFS for children, adolescents, and young adults was 58%, 41%, and 24%, respectively, and the 5-year OS was 72%, 65%, and 25%, respectively [14]. Among patients > 16 years of age with Ph-like B-ALL treated on the Dutch–Belgian HOVON (Stichting Hemato-Oncologie voor Volwassenen Nederland) study group clinical trials from 1993–2009, more than half (57%) of those who were subsequently identified as Ph-like ALL patients were categorized as standard risk at the time of diagnosis and only 71% attained complete remission (CR) following front-line treatment. The 5-year cumulative incidence of relapse, EFS, and OS were 67%, 19%, and 28%, respectively [13]. A single-institution analysis from the MD Anderson Cancer Center of 56 patients >15 years of age with Ph-like B-ALL also revealed poor outcomes [16]. Although over 90% of these patients achieved complete remission, only 30% achieved an MRD-negative remission. MRD negativity had no impact on outcomes with the median OS being 26 months in the MRD-negative group compared to 23 months in the MRD-positive group (*p* = 0.318). The 5-year OS was 23% in the Ph-like B-ALL group compared to 59% in the B-other group (*p* = 0.006). Finally, among AYAs 18–40 years treated on the US Intergroup C10403 trial, 2-year EFS was 52% versus 81% (*p* = 0.04) [4] and 3-year overall survival (OS) was 63% versus 81% [2] for patients with and without the Ph-like signature, respectively. In a Japanese cohort of pediatric B-ALL patients harboring Ph-like kinase fusion genes, the 5-year event-free and overall survival rates were 48% and 73%, respectively, underscoring the inferior outcomes in this subset across the globe [10]. Despite the inferior outcomes, when using the traditional ALL criteria for risk stratification (e.g., age, White Blood Cell (WBC) count, high-risk cytogenetics) at least half of Ph-like B-ALL patients would fall under low or standard risk groups, underscoring the need to incorporate this feature as a separate poor risk factor [32]. 

While some studies indicate that MRD-directed therapy can improve outcomes in these patients [10,24], other studies demonstrate otherwise [2,24,25,33]. Data from a limited number of patients derived from the HOVON study group clinical trials indicate that allogeneic stem cell transplant (ASCT) in the first complete remission may improve outcomes in Ph-like B-ALL. Only one relapse was seen among five patients who underwent ASCT, whereas nine of the 15 patients treated with chemotherapy alone relapsed [13]. Nevertheless, additional therapies to target the unique biological aberrations of Ph-like ALL are necessary, in addition to MRD-directed therapy, to improve outcomes in this high-risk ALL subgroup. 

## 4. Role of Lymphoid Transcription Factors IKAROS and PAX5

IKAROS is a zinc finger protein that binds to and regulates DNA transcription [34,35]. It has recently been implicated as an epigenetic regulator [36] with the ability to induce active enhancers and super-enhancers [37]. IKAROS plays key roles in hematopoiesis and lymphopoiesis [38,39] and is of importance for B-ALL as it is essential for the differentiation of B cell precursors [40]. 

IKAROS is encoded by the *IKZF1* gene which is deleted in 60–80% of Ph-positive ALL [17,41] and is associated with failure of tyrosine kinase inhibitor (TKI) therapy [41]. *IKZF1* deletion has been reported in 80% [17] of Ph-like B-ALL and contributes to the poor prognosis of this disease entity [12,21,31]. Several types of *IKZF1* aberrancies have been observed in Ph-like B-ALL [21] including deletion of the entire locus, of subgroups of exons, or of genes upstream. Deletion of exons 3 through 6 results in the expression of a dominant negative isoform, which lacks the N-terminal DNA binding domain but retains the C-terminal dimerization domain. This short isoform binds to other IKAROS proteins, sequesters them from DNA, and thereby inhibits their function [42]. Although the extent of IKAROS tumor suppressor mechanisms are unclear, it has been shown to bind DNA and regulate the expression of genes involved in cellular proliferation [43]. Further, the loss of IKAROS regulatory activity in B cell precursors is believed to contribute to the block in B cell differentiation that is characteristic of B-ALL cells [40]. *IKZF1* deletion also contributes to chemoresistance in the leukemic cells [12].

A genetic aberration in *IKZF1* alone may be insufficient for poor outcomes, which likely requires a synergistic effect with other molecular factors. The oncogenic kinase Casein Kinase 2 (CK2) is over-expressed in B-ALL [43]. CK2 impairs the ability of Ikaros to bind DNA and exert its tumor suppressor function [44,45,46]. Song et al. [43] demonstrated the antileukemic effects of CK2 inhibitors in patient derived xenograft models of Ph-like ALL where CK2 inhibition increased the activity of IKAROS protein produced in cells with monoallelic *IKZF1* deletion.

The *PAX5* gene encodes for the PAX5 (paired-box 5) protein which is also a DNA binding transcription factor [47] essential for B cell precursor differentiation [48]. *PAX5* is an important tumor suppression gene in ALL and is altered in a third of Ph-like B-ALL cases [12,21]. Mutations in *PAX5* haplo-insufficient mice increase the penetrance and reduce the latency of development of B-ALL [49]. *IKZF1* and *PAX5* alterations frequently occur together [50,51]. While only *IKZF1*, and not *PAX5* alterations, was independently associated with poor outcomes in one large study, defects in multiple B cell differentiation regulators (such as PAX5 in combination with others) were also associated with worse outcomes [21]. 

## 5. Role of CRLF2

CRLF2 (Cytokine receptor-like factor 2) [52] partners with IL7Rα to form a heterodimeric receptor complex that binds the ligand thymic stromal lymphopoietin receptor (TSLP) [53,54,55]. This ligand–receptor interaction leads to downstream activation of the JAK2/STAT5 and the PI3K/AKT/mTOR pathways [56,57,58]. Figure 1 demonstrates the CRLF2 receptor complex and downstream pathways. Rearrangement of the *CRLF2* gene located on Xp22.3 and Yp11.3, with the resulting overexpression of the CRLF2 receptor complex is seen in 10–15% [28,30] of all high-risk B-ALL. Among Ph-like B-ALL, the prevalence of CRLF2 rearrangement can be as low as 24% in children with NCI-standard risk disease to as high as 60% in adolescents [9,14]. It was first described by Russell et al. in 2009 [59] in a UK cohort of B-ALL and subsequently by Harvey et al. in a US cohort [30]. 

Two types of *CRLF2* rearrangements have been reported. *IgH-CRLF2* involves translocation of the immunoglobulin heavy chain gene *IGH*@ on 14q32 to *CRLF2* in the pseudo-autosomal region 1 (*PAR1*) of Xp22.3/Yp11.3. *P2RY8-CRLF2* involves fusion due to an interstitial deletion of the pseudo-autosomal region 1 (*PAR1*) centromeric of *CRLF2* in chromosomes X and Y. The latter is the predominant abnormality in Down syndrome associated B-ALL (DS-ALL) being present in ~50% of such cases [11]. Patients with *IgH-CRLF2* tend to be older (median age of 16 years), whereas those with *P2RY8-CRLF2* tend to be younger with a median age of 4 years [59]. In a single institution retrospective analysis of patients >15 years with Ph-like B-ALL, 66% were CRLF2 overexpressing. Among them, 76% had *IgH-CRLF2*, whereas only 17% had *P2RY8-CRLF2* [16]. JAK mutations are present in more than 50% of *CRLF2* rearranged Ph-like ALL [9,14,16,30,32,58,60].

*CRLF2* gene rearrangement and receptor overexpression is strongly associated with Hispanic/Latino ethnicity in both children and adults. *CRLF2* alterations occur in 35% of Hispanic/Latino pediatric B-ALL patients as opposed to only 11% among non-Hispanic pediatric B-ALL patients [30]. In adults, occurrence of the *CRLF2* rearrangement in B-ALL is approximately 27% overall, but up to 45% in Hispanic patients [61]. The inherited *GATA3* variant rs3824662 has been shown to have an association with Ph-like ALL with *CRLF2* rearrangement, *JAK* mutation, *IKZF1* deletion, and increased risk of relapse [62]. Among patients with Ph-like ALL, this allele has been described in ~50% of Native American Guatemalans, 40% of US Hispanics, and only 14% of Europeans and provides a biologic basis for the racial disparities in ALL incidence [63] and outcomes [62,64].

*CRLF2* rearrangement and overexpression predicts poor outcomes with 4-year relapse-free survival of 35% with *CRLF2* versus 71% without *CRLF2* alterations [28,30,59]. *CRLF2* overexpression, *JAK* mutations, and *IKZF1* deletions orchestrate leukemogenesis and contribute to dismal prognosis with survival rates of 25–30%. *CRLF2-*overexpressing B-ALL-harboring *IKZF1* deletion is associated with increased risk of relapse even with low MRD levels [33]. Therefore, risk adapted therapy using MRD alone may not be sufficient in reducing relapse in these patients, underscoring the need for novel targeted therapies [2]. 

IKAROS protein has been shown to directly bind to the *CRLF2* promoter and regulate CRLF2 expression in leukemic cells. CK2 (Casein kinase 2) inhibitor has been shown to increase IKAROS activity and binding to the *CRLF2* promoter and to suppress CRLF2 expression [29]. Preclinical in vivo models of *CRLF2* B-ALL are challenging due to the low homology between mouse and human *CRLF2* and its ligand, TSLP. Patient-derived xenograft (PDX) models are even more challenging since mouse TSLP does not activate the human TSLP receptor complex. A novel PDX model engineered by Francis et al. [65] provides human TSLP and thus provides an important preclinical model for understanding disease mechanisms and identifying therapies that can be effective against Ph-like B-ALL with *CRLF2* rearrangements. 

## 6. Kinase Activating Alterations in Ph-like B-ALL

Roberts et al. [14] reported on the kinase activating genetic alterations in Ph-like ALL that were noted in 10% of children and 27% of young adults with B-ALL and mirrored the poor outcomes seen by other groups. Kinase activating gene alterations were noted in 91% of patients with Ph-like B-ALL and included ABL fusions (*ABL1*, *ABL2*, *CSF1R*, *PDGFRB*) in 12%, *EPOR* rearrangement in 4%, *JAK2* rearrangement in 7%, *CRLF2* rearrangement in 50%, other JAK/STAT alterations (*IL7R*, *FLT3*, *SH2B3*, *JAK1*, *JAK3*, *TYK2*, *IL2RB*) in 12%, *RAS* mutations in 4%, and uncommon fusions (*NTRK3* or *DGKH*) in 1%. These aberrations, depicted in Figure 2, lead to phosphorylation and activation of STAT5 and cytokine independent proliferation. *ABL* rearrangements were noted to be more common in children while *JAK2* rearrangements were more common in young adults. Druggable targets were identified in vitro with *ABL1*, *ABL2*, *CSF1R*, and *PDGFRB* fusions sensitive to Dasatinib, *EPOR* and *JAK2* rearrangements sensitive to JAK inhibitor Ruxolitinib, and ETV6-NTRK3 fusion responding to Crizotinib. Other studies have shown efficacy of Ruxolitinib and Rapamycin (mTOR inhibitors) in *CRLF2* and *JAK* aberrant B-ALL xenografts [58,66].

Reshmi et al. identified the Ph-like gene signature in 20% of children from the COG AALL1131 trial [32]. Among them, *CRLF2* rearrangement was noted in 43%, with half of those harboring mutations in the JAK/STAT pathway (*JAK1*, *JAK2*, *IL7R*). *ABL* class fusions (*ABL1*, *ABL2*, *CSF1R*, *PDGFRB*) were present in 14%, EPOR rearrangements or JAK2 fusions in ~9%, other JAK/STAT alterations (*IL7R*, *SH2B3*, *JAK1*) in 6%, other kinases (*FLT3*, *NTRK3*, *LYN*) in ~5%, and RAS pathway mutations (*KRAS*, *NRAS*, *NF1*, *PTPN11*) in 6%. Figure 3 depicts the above alterations.

Multiple tyrosine kinase fusion genes were also identified by Boer et al. in bcr-abl1-like ALL [68]. These include *EBF1-PDGFRB*, *SSPB2-CSF1R*, *ZM1Z1-ABL1*, *FOXP1-ABL1*, *RCSD1-ABL2*, *TERF2-JAK2*, *BCR-JAK2*, and *PAX5-JAK2*. Sequence mutations involving *IL7R*, *KRAS* and *NRAS* are seen in ~5% of Ph-like ALL [9]. Tyrosine kinase fusions and *JAK2* mutations are mutually exclusive, the latter occurring only in *CRLF2*-overexpressing cases. Tyrosine kinase fusion genes are also commonly associated with *IKZF1* deletions and poor prognosis. 

*EPOR* (erythropoietin receptor) rearrangements in Ph-like ALL were further detailed by Iacobucci et al. [69] and involved insertion of the *EPOR* gene (located on chromosome 19) adjacent to the *IGH* or *IGK* loci, a t(14;19) that places *IGH* in proximity to *EPOR*, or an intrachromosomal inversion of chromosome 19 that juxtaposes *EPOR* to *LAIR1* (leucocyte-associated IG-like receptor 1). These changes lead to truncation of the cytoplasmic tail of EPOR, loss of regulatory residues, increased sensitivity to ligand binding by erythropoietin, and JAK/STAT activation. *EPOR* rearranged B-ALL was sensitive to JAK inhibition (Ruxolitinib) in vitro [69]. *ETV6-ABL1* fusion occurs in <1% of childhood and adult ALL cases. Eighty percent of these cases harbor concurrent *CDKN2A/B* and *IKZF1* deletions and have poor outcomes with <40% overall survival, similar to other bcr-abl-like ALL [70]. 

## 7. Diagnosis of Ph-like ALL: Screening and Subtype Confirmation

The multiplicity of genetic and molecular aberrations that characterize Ph-like ALL has made it difficult to implement a single set of tests for screening and subtype confirmation that can be used across all settings. Testing can therefore be comprehensive, utilizing whole genome, whole exome, or transcriptome sequencing, or limited to identifying the major subtypes only. The former methods are utilized by a few major research institutions whereas the latter are offered by several commercial and institutional laboratories. An algorithmic approach to the diagnosis of Ph-like ALL is summarized in Figure 4. 

### 7.1. Gene Expression Profiling/Analysis

Gene expression analysis to classify Ph-like ALL was first reported by Den Boer et al. in the Netherlands and Mullighan et al. in the US [12,21]. The former group applied hierarchical clustering using 110 gene probe sets and classified pediatric ALL into six genetic subtypes, one of which was bcr-abl-like. The latter group utilized a 257 gene probe set and a prediction analysis of microarrays to identify Ph-like ALL. A risk score was developed based on supervised principal-components analysis and identified an association between copy number status of 20 genes and treatment outcome. 

A 15 gene (*IGJ*, *SPATS2L*, *MUC4*, *CRLF2*, *CA6*, *NRXN3*, *BMPR1B*, *GPR110*, *CHN2*, *SEMA6A*, *PON2*, *SLC2A5*, *S100Z*, *TP53INP1*, *IFITM1*) low density array (LDA) classifier developed by Harvey et al. [71] was able to predict Ph-like ALL with high sensitivity (93%) and specificity (89%). The same group also developed an 8-gene assay (*JCHAIN*, *SPATS2L*, *CA6*, *NRXN3*, *MUC4*, *CRLF2*, *ADGRF1*, *BMPR1B*) to identify Ph-like ALL with targetable kinase gene aberrations [32] and generated an integrated score between 0 and 1, with a score of ≥ 0.5 being predictive of Ph-like ALL. LDA can be used as a screening test to identify Ph-like ALL and further characterization can be done with FISH, RT-PCR, and flow cytometry. 

### 7.2. Fluorescent in Situ Hybridization (FISH)

The majority of chromosomal rearrangements in Ph-like ALL are cytogenetically cryptic, that is, they cannot be detected by conventional karyotyping. Many labs throughout several countries now offer FISH analysis to detect the commonly known gene translocations. These include *ABL1*, *ABL2*, *CRLF2*, *JAK2*, *EPOR*, *PDGFRB*, and *CSF1R*. 

### 7.3. Reverse-Transcriptase Polymerase Chain Reaction (RT-PCR)

RT-PCR is a relatively rapid method of confirming the same gene abnormalities detected by gene expression profiling by RNA sequencing. Chiaretti et al. developed a quantitative RT-PCR-based model to predict bcr/abl1-like cases by using a 10 (overexpressed) gene signature (*SOCS2*, *IFITM1*, *CD99*, *TP53INP1*, *IFITm2*, *JCHAIN*, *NUDT4*, *ADGRE5*, *SEMA6A*, *CRLF2*) [26].

### 7.4. Flow Cytometry

CRLF2 (TSLPR) is overexpressed in approximately half of Ph-like ALL cases and can be quickly identified by flow cytometry. CRLF2 flow cytometry is part of the standard Ph-like panel of many commercial and institutional laboratories. 

### 7.5. Next-Generation Sequencing (NGS)

Extensive gene sequencing can be done with transcriptome sequencing, whole genome sequencing, or whole exome sequencing to identify Ph-like ALL, including the aberrations that may be missed by the 8- or 15-gene LDA analyses [14]. Transcriptome sequencing involves extensive RNA sequencing. Whole exome sequencing (WES) is limited to the coding regions and therefore misses anomalies that may occur in non-coding regions, for example gene rearrangements involving non-coding regions. Whole genome sequencing (WGS) can capture abnormalities in coding as well as non-coding regions. All three sequencing methods are expensive, time consuming, and used only in a few large research institutions. 

## 8. Therapeutic Avenues

Since multiple studies have demonstrated poor outcomes with pediatric and adult treatment regimens with and without allogeneic stem cell transplant in Ph-like ALL across all age groups [4,9,10,11,12,13,15,16,17,21,25,27,28,30,32,33,41,62,67,72], there is an unmet need for novel therapies. The plethora of genomic and molecular alterations in Ph-like ALL opens up multiple avenues for targeted therapies. These include, but are not limited to, JAK (Ruxolitinib), ABL (Dasatinib), mTOR and other kinase pathway inhibitors, and immune therapies against surface molecules. Table 2 summarizes preclinical studies and Table 3 summarizes clinical studies using targeted therapies. 

### 8.1. JAK/STAT Pathway Targets

*JAK* aberrations are present in approximately 40% of all Ph-like ALL, primarily in the *CRLF2* subgroup where about half the cases have *JAK* mutations [14,32,58,72]. This leads to activation of the JAK/STAT/PI3K/mTOR pathway [65] and opens up opportunities for the use of JAK, mTOR, and PI3K inhibitors alone or in combination as shown *in vitro* and *in vivo* preclinical studies [58,66,73]. The combination of JAK inhibition (with Ruxolitinib) and MEK inhibition (with Selumetinib) was also shown to have *in vitro* and *in vivo* efficacy in PDX models of *JAK* mutated B-ALL [74]. Currently, at least three clinical trials are underway to investigate the addition of Ruxolitinib to combination chemotherapy in *CRLF2* rearranged/*JAK* aberrant Ph-like B-ALL (NCT02723994, NCT03117751, NCT02420717). Total Therapy XVII for Newly Diagnosed Patients with ALL (NCT03117751) is a phase 2/3 clinical trial designed for early identification of patients with targetable lesions and addition of Tyrosine Kinase inhibitors (TKIs) to the chemotherapy backbone. Patients with JAK/STAT signaling pathway activation receive Ruxolitinib and those with *ABL* class fusions receive Dasatinib. The MD Anderson Cancer Center is evaluating Ruxolitinib or Dasatinib with chemotherapy in treating adult patients with relapsed or refractory Ph-like ALL (NCT02420717)**.** Additionally, investigators at Oregon Health Sciences University are leading a study exploring personalized kinase inhibitors in combination with chemotherapy for patients with newly diagnosed acute myeloid leukemia (AML) or ALL. Based on the results of *in vitro* kinase inhibitor assay, patients in the ALL cohort receive Dasatinib, Ponatinib, or Ruxolitinib (NCT02779283).

Dose finding results from the Phase 2 study of Ruxolitinib with chemotherapy in 40 children and AYAs with *CRLF2* or other *JAK* aberrant, newly diagnosed, high-risk Ph-like ALL were reported in 2018 [75]. Ruxolitinib was added to consolidation chemotherapy with an augmented Berlin-Franfurt-Munster (BFM) protocol on the AALL/1521 study. Cytopenias (>60%), febrile neutropenia (~70%), and transaminitis (~50%) were commonly observed. Eighty percent of patients experienced grade 3 or 4 events related to Ruxolitinib without identified dose-limiting toxicities. Plasma drug levels at 4 hours were consistent with the known pharmacokinetic profile of Ruxolitinib, and pharmacodynamic studies demonstrated inhibition of phosphorylated STAT5. 

### 8.2. mTOR Inhibition

mTOR inhibition alone was shown to decrease leukemia growth in a preclinical model of high-risk ALL characterized by early relapse [77]. Combining mTOR inhibition with histone deacetylase inhibition was shown to have activity in Ph+ and Ph-negative cell lines [76]. Similarly, combining Dasatinib with mTOR inhibitor was more effective than Dasatinib alone in a preclinical model of Ph-like ALL harboring *ABL1* or *ABL2* rearrangements [88]. A phase 1 trial of the mTOR inhibitor, Temsirolimus, with intensive re-induction for relapsed ALL (COG study ADVL1114) induced remission in 50% of the patients. However, it resulted in excessive toxicity and was therefore not tolerable in these heavily pretreated children [89]. A potential strategy to overcome this drawback is to identify mTOR-responsive leukemias at diagnosis and incorporate lower doses of an mTOR inhibitor staggered to the chemotherapy backbone. 

### 8.3. ABL Inhibition

*ABL* class fusions, involving *ABL1*, *ABL2*, *CSF1R*, and *PDGRFB*, are present in approximately 10% of Ph-like B-ALL and are targetable with ABL tyrosine kinase inhibitors such as Dasatinib [14,32,83,88]. Clinical trials testing Dasatinib in patients harboring these lesions are also using Ruxolitinib if there is JAK/STAT signaling pathway activation and are detailed in the section above titled JAK/STAT pathway targets.

### 8.4. Other Therapeutic Avenues

Qin et al. reported on the preclinical efficacy of chimeric antigen receptor expressing T cells against TSLPR (CRLF2) [90]. Birinapant, a SMAC (second mitochondria-derived activator of caspases) mimetic, has shown preclinical activity against Ph-like B-ALL xenografts. SMAC is an endogenous antagonist of IAPs (inhibitors of apoptosis) and is released from mitochondria in response to apoptotic stimuli with subsequent activation of downstream caspases [91]. 

### 8.5. Other Molecular Aberrations/Targets

LNK (*SH2B3*), a lymphocyte adaptor protein, inhibits cytokine signaling during hematopoietic stem cell renewal and B cell development. *LNK* mutations are seen in ~6% of pediatric and young adult Ph-like ALL. Co-absence of *TP53* and *LNK* genes have been shown to initiate pre-B ALL that resembles Ph-like B-ALL with JAK/STAT/mTOR pathway activation and responds in vitro and in vivo to JAK inhibitor (Ruxolitinib) and the dual mTOR and PI3K inhibitor (BEZ235) [85]. Ge et al. identified a high-risk subset of B-ALL that exhibited high IL7R and low SH2B3 expression that was associated with *IKZF1* deletion. Use of a Casein Kinase 2 inhibitor was shown to restore IKAROS function in these cells [92]. 

## 9. Summary

Ph-like ALL is a high-risk subgroup of ALL with poor clinical outcomes across all age groups. It is characterized by a heterogenous genetic and molecular landscape populated by *IKZF1* deletion, *CRLF2* rearrangement and overexpression, *JAK* mutations, *ABL* class fusions, *EPOR* rearrangements, tyrosine kinase pathway activation, and several other less common aberrations. The biologic heterogeneity of this disease entity leads to challenges not only in diagnosis but also in the investigation of novel therapies. Several diagnostic laboratory tests are available. However, unlike Ph-positive ALL, there is no uniformity in the diagnostic methodology employed. Several diagnostic panels are employed by different commercial and institutional laboratories. In this review, we have suggested an algorithm for the laboratory diagnosis of Ph-like ALL in the clinical setting (Figure 4). 

Although the outcomes are poor, there is an immense potential for targeted therapeutic strategies involving kinase inhibitors as well as immune therapies targeting the CRLF2 molecule on the cell surface. To this effect, there are several ongoing clinical trials testing kinase inhibitors and several more preclinical studies investigating novel therapies. Current clinical trials utilize primarily JAK and ABL kinase inhibitors (Ruxolitinib and Dasatinib). Future trials should incorporate other agents targeting many other genomic/molecular aberrations. Until then, early identification, referral to a clinical trial, MRD-directed therapeutic strategies incorporating CD19 and CD22 targeted agents (Blinatumumab and Inotuzumab, respectively) and chimeric antigen receptor cell therapies, and early referral for hematopoietic stem cell transplant may improve outcomes. 

Taken together, what is currently a difficult leukemia to cure will hopefully evolve into a highly curable disease thanks to the efforts of several investigator groups worldwide. 

## Figures and Tables

**Figure 1 ijms-21-02193-f001:**
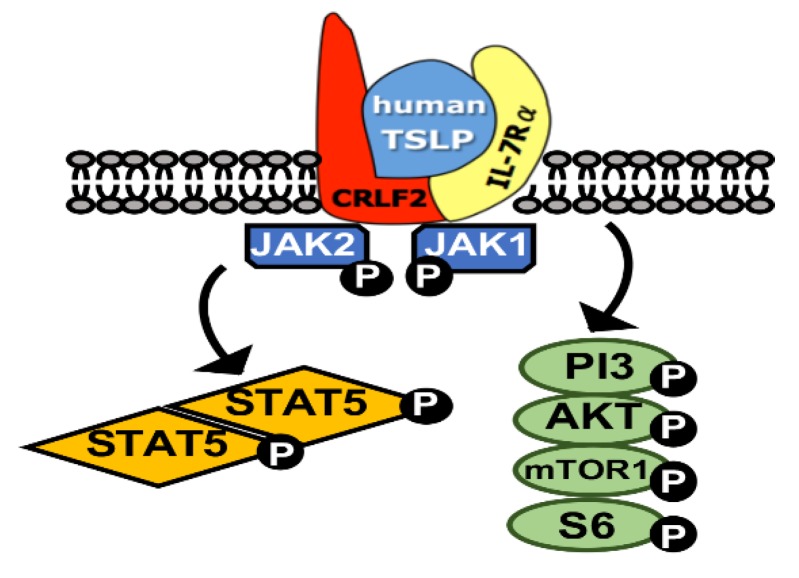
Schematic representation of the CRLF2-receptor complex.

**Figure 2 ijms-21-02193-f002:**
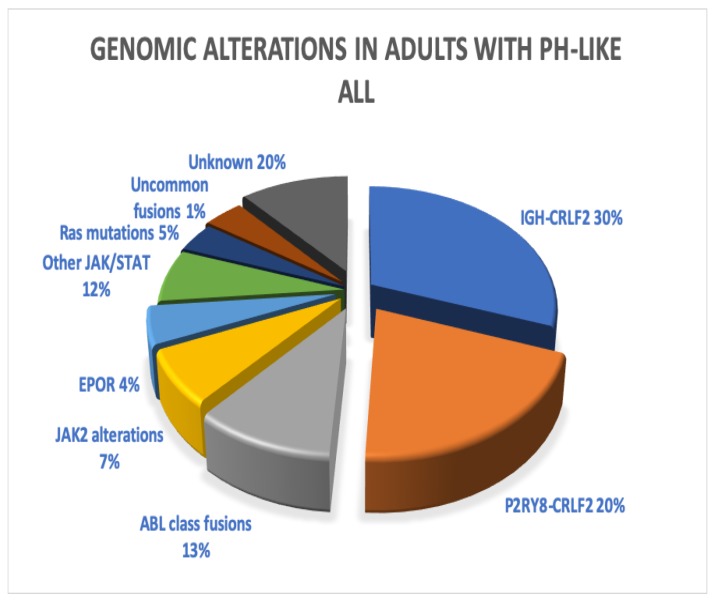
Genomic alterations in adults with Ph-like acute lymphoblastic leukemia (ALL). Pie chart constructed with data from Roberts et al. [67].

**Figure 3 ijms-21-02193-f003:**
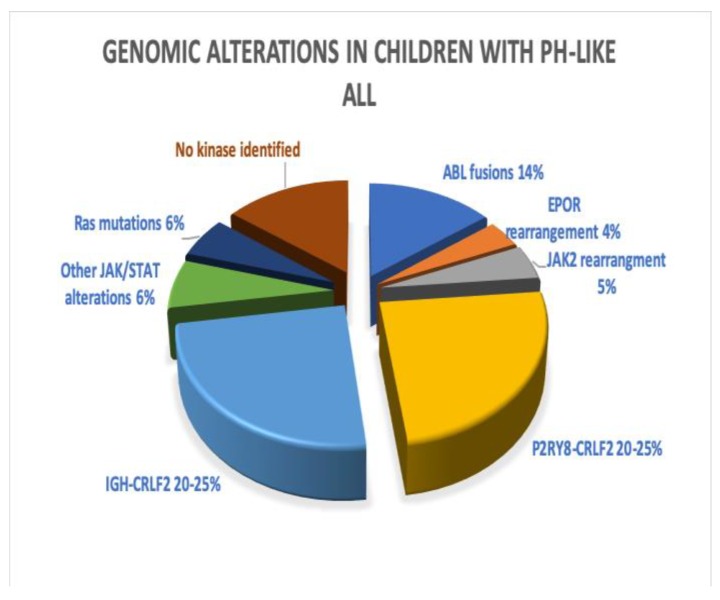
Genomic alterations in children with Ph-like ALL. Pie chart constructed with data from Roberts et al. and Reshmi et al. [14,32].

**Figure 4 ijms-21-02193-f004:**
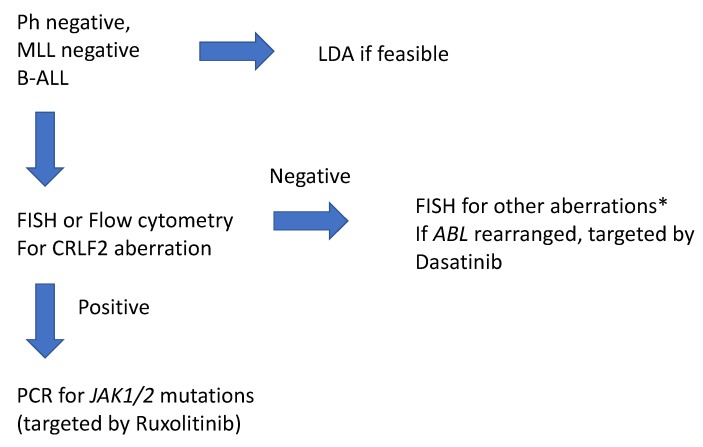
Proposed testing algorithm for Ph-like ALL in the clinical setting. MLL: Mixed Lineage Leukemia; FISH: fluorescent in situ hybridization; PCR: polymerase chain reaction; LDA: low density array. *Other aberrations include—ABL1, ABL2, CRLF2, JAK2, EPOR, PDGFRB, and CSF1R.

**Table 1 ijms-21-02193-t001:** Differences between the cohorts and methodologies used in the initial description of Philadelphia (Ph)-like B-ALL [12,21].

Feature	US COG Cohort [21]	DCOG Cohort [12]
Number of genes probed	257 gene probe sets	110 gene probe sets
Methodology	Prediction analysis of microarrays	Hierarchical clustering
Patient characteristics	High-risk patients (CNS or testicular disease, MLL rearrangement, age >10 years, male sex and WBC > 50,000) in the original cohort and all risk groups in the validation cohort	All risk groups
Racial/ethnic differences	Presence of Hispanic/Latino ethnicity	Lack of Hispanic/Latino patients
*CRLF2* aberration/*JAK2* mutation	Higher frequency	Lower frequency
Outcomes	Poor outcomes in IKZF1-deleted or mutated cases	Six genetic subtypes of ALL identified BCR/ABL1-like had poor outcomes compared to other groups

**Table 2 ijms-21-02193-t002:** Preclinical studies identifying kinase mutations and targeting with kinase inhibitors in Ph-like B-ALL.

Kinase Target	Inhibitor	Reference
JAK, ABL, *ETV6-NTRK3* fusion	Ruxolitinib, Dasatinib, Crizotinib	Roberts et al.; 2014 [14]
JAK+MEK	AZD1480+Selumetinib	Suryani et al.; 2015 [74]
mTOR+HDAC	mTOR inh+Vorinostat	Beagle et al.; 2015 [76]
mTOR	mTOR inhibitor	Hasan et al.; 2015 [77]
*RCSD1-ABL1* rearr. t(1;9)	ABL inhibitor Ponatinib	Collette et al.; 2015 [78]
HSP90/JAK	HSP90 inhibitor	Kucine et al.; 2015 [79]
*ABL1* (*NUP214-abl1* fusion)	Dasatinib	Duployez et at; 2016 [80]
SYK (spleen tyrosine kinase)	PRT318/R406 (Fostamatinib)	Kohrer et al.; 2016 [81]
*ABL1*(*RCSD1-ABL1*)	Imatinib	Perwein et al.; 2016 [82]
*ATF71P/PDGFRB*	Dasatinib	Kobayashi et al.; 2014 [83]
*EBF1-PDGFRB* fusion	Imatinib	Weston et al.; 2013 [84]
Lnk negative/ p53 negative blasts	Ruxolitinib/BEZ235 (dual mTOR and PI3K)	Cheng et al.; 2016 [85]
C-myc and JAK2	JQ1 (BET bromodomain) and Ruxolitinib	Kim et al.; 2018 [86]
SRC/ABL	Dasatinib	Sarno et al.; 2018 [87]

**Table 3 ijms-21-02193-t003:** Ongoing clinical trials of Ph-like ALL (Clinicaltrials.gov).

Sponsor/Collaborator	Trial	NCT#
Children’s Oncology Group	Phase 2 study of Ruxolitinib with chemotherapy for CRLF2- or JAK-mutated B-ALL	NCT02723994
University of New Mexico	ALL therapies informed by genomic analysis	Single arm, open label NCT02580981
University of Chicago	Phase 1 trial of Ruxolitinib in combination with pediatric-based regimen for AYAs with Ph-like ALL.	NCT03571321 (not yet recruiting as of 12/2019)
MD Anderson Cancer Center	Ruxolitinib or Dasatinib with chemotherapy in relapsed/refractory Ph-like ALL	NCT02420717
St. Jude’s Children’s Research Hospital	Total therapy XVII: Dasatinib for ABL1 class fusion and Ruxolitinib for activated JAK/STAT signaling	NCT03117751
OHSU Knight Cancer Institute	Personalized kinase-inhibitor therapy combined with chemotherapy	NCT02779283
University of Washington	In vitro-sensitivity directed therapy	NCT02551718
St. Jude’s Children’s Research Hospital	Treatment with combination chemotherapy for relapsed or refractory ALL	NCT03515200
Nanfang Hospital of Southern Medical University, China	To evaluate safety and efficacy of adding Chidamide (oral histone deacetylase inhibitor) and Dasatinib (TKI) to pediatric-inspired and MRD-directed pediatric protocol for Ph-like ALL	Open label, two-arm trial

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
