# Peer review of "The Current Genomic and Molecular Landscape of Philadelphia-like Acute Lymphoblastic Leukemia"

_ijms, 2020, doi:10.3390/ijms21062193_

Round 1

Reviewer 1 Report

The manuscript by Shiraz et al “The Current Genomic and Molecular Landscape of Philadelphia-like Acute Lymphoblastic Leukemia” summarized the current geneomic and signalling pathways associated with Philadelphia-like Acute Lymphoblastic Leukemia and highlighted the current therapies for this disease. The manuscript is clearly written. I have only a few minor comments.

  1. Line 46: “However, it was not until 2009 when the molecular and clinical description of Ph-like B-ALL was reported simultaneously by two independent groups of investigators, St. Jude’s/Children’s Oncology Group (COG) in the United States (U.S.) and the Dutch Childhood Oncology Group (DCOG), based on the similar gene expression profile as Ph-positive ALL but without the hallmark BCR-ABL1 translocation.” This is quite long sentence.  If possible, please split into two sentences.

  1. Line 139 “In those studies CK2 inhibition increases the activity of the Ikaros protein produced in cells with monoallelic IKZF1 deletion [45]”. The author only cited one references, please include other references here as well.

  1. Line 174: “ALL. [30].”. The author should correct to “ALL [30]”.

  1. Figure 1, 2, 3 were not mentioned in the text. So, they should be mentioned somewhere in the text.

  1. Chapter 3 is quite long. I would suggest to split into 2 more chapters. For examples, chapter 4 “The current therapeutic avenues for Ph-like B-ALL” (from line 245 to line 313).
  2. The supporting diagrams (if possible) to summarize all signaling of genomic alterations in Ph-like ALL and how current drugs targeting these pathways would definitely enhance the quality of this article.

Generally, these are all minor comments and overall this is an interesting review.

Author Response

  1. Line 46 has been split into 2 sentences as suggested. 
  2. Line 139 (now 140) has only one reference (44) and sentence has been corrected to reflect that. 
  3. Line 174 (now 175) has been corrected as suggested. 
  4. Figures 1, 2, and 3 are now referred in the text. 
  5. The chapter on therapeutic avenues has been split based on drug targets, and whether preclinical data or clinical trials. 
  6. I added a figure with an algorithmic approach to diagnosis and included drug targets. 

Thank you for the suggestions. These have definitely improved the quality of this review. 

Reviewer 2 Report

It is well written review. But there are points to improve in some sections.

Figures 2 and Figure 3: Referring the original figures are fine, but they can be copyright violations, which must be cleared from the publisher

For Diagnosis in Clinical setting; This part needs improvement. Flow cytometry for CRLF2 must be discussed. It is ~50% of adult cases. The list of labs for FISH panels are not complete (as far I know FISH panels in MD Anderson not included; there may be other missing academic laboratories) and the list probably does not serve much for the paper anyways, as this is a scientific review not infomercial which tests are available in which US clinical laboratories. PCR based testing (one lab I know offering test is Nationwide Childrens in Columbus, OH) should also be the part for laboratory diagnosis in a clinical setting and research applications such as WGS or Optical mapping must be discussed.

For other sections: No points to improve.

Author Response

  1. Figures 2 and 3 (pie charts) were constructed by the first author with data from the references. They were not copy/pasted from the reference. I have modified the description to reflect that. 
  2. The section describing diagnosis has been completely changed to reflect the reviewers suggestions. 

Thank you for the suggestions. These have definitely made this article much better. 

Reviewer 3 Report

Overall, the topic is worthwhile. People can catch the significant knowledge base of this topic by reading this manuscript. It covers the history of the discovery and the current situation from benchside to bedside. Tables (2, 3, and 4) are clear to read as good summaries.

However, to section the paragraphs, following the section-3 "clinical features and outcomes", are there missing numbers to order the remaining parts? Or are these parts after section-3 all subtitles to any part? Are they parallel?

The "summary" section looks not very comprehensive. Is there still the complex nature of this disease and any challenge that future study needs to overcome?

For table-1, since the author claimed that "although their methodologies differed, both groups demonstrated the complexity and poor clinical outcomes". The comparison of the methodologies may be less meaningful than the comparison of the actual complexity and the poor clinical outcomes between the two independent discoveries of the two groups.

In table-1, as it is claimed the search is for "other genetic alterations"(without BCR-ABL1 translocation), why is the gene set selected to "identify BCR-ABL1 positive cases"?

In Figures 2 and 3, even though adapted, please make sure not to copy and paste directly. Make it in your own way.

Author Response

  1. The numbering of the paragraphs have been removed to avoid confusion. Thank you for pointing that out. 
  2. Summary section - I added more data points as suggested to try and make the summary section more comprehensive. 
  3. Table 1 has been changed to reflect outcomes and some non-relevant methodology comparison has been removed. 
  4. Table 1 - row depicting "gene set selected to identify bar-abl1 positive cases" has been removed to avoid confusion. 
  5. Figures 2 and 3 were constructed by the first author using the data from the references noted and were not copy/pasted. The figure description has now been changed to reflect that. 

Thank you for the valuable suggestions. They have definitely improved the quality of the manuscript. 

Round 2

Reviewer 3 Report

Well done improvement based on the reviewers' comments.